# Analysis of Atom-level pretraining with QM data for Graph Neural Networks Molecular property models

Jose A. Arjona-Medina [* 1]   Ramil Nugmanov [* 2]

## Abstract

Despite the rapid and significant advancements in deep learning for Quantitative Structure-Activity Relationship (QSAR) models, the challenge of learning robust molecular representations that effectively generalize in real-world scenarios to novel compounds remains an elusive and unresolved task. This study examines how atom-level pretraining with quantum mechanics (QM) data can mitigate violations of assumptions regarding the distributional similarity between training and test data and therefore improve performance and generalization in downstream tasks. In the public dataset Therapeutics Data Commons (TDC), we show how pretraining on atom-level QM improves performance overall and makes the activation of the features distributes more Gaussian-like which results in a representation that is more robust to distribution shifts. To the best of our knowledge, this is the first time that hidden state molecular representations are analyzed to compare the effects of molecule-level and atom-level pretraining on QM data.

## 1. Introduction

Despite the rapid and significant advancements in deep learning for Quantitative Structure-Activity Relationship (QSAR) models, the challenge of learning robust molecular representations that effectively generalize in real-world scenarios to novel compounds remains an elusive and unresolved task.

At the core of supervised learning with deep neural networks, there is the underlying assumption that the function we aim to approximate is "smooth" enough, meaning that small changes in the input should not result in large changes

in the output (Goodfellow et al., 2016). However, in the application field of QSAR, we often experience regions where small chemical modifications drastically change the biological response (Cruz-Monteagudo et al., 2014). For example, the extreme forms of Structure-Activity Relationship (SAR) discontinuity are called activity cliffs (AC) (Bajorath et al., 2009), which are formed by pairs of structurally similar compounds with large differences in potency [(Stumpfe & Bajorath, 2012),(Peltason & Bajorath, 2011)]. One way of overcoming this problem would be to increase the number of datapoints since the smoothness assumption works well as long as there are enough examples for the learning algorithm to observe high points on most peaks and low points on most valleys of the true underlying function to be learned (Goodfellow et al., 2016). However, it is not a general or scalable solution because the costs of producing additional experimental datapoints are high and would be applicable only locally for a specific target property

Supervised Learning methods also assume that test data comes from the same underlying distribution as training data (Bishop, 2006). Although novel techniques often present impressive metrics, most often they do not suffice to meet the practical needs in real-world drug discovery (Deng et al., 2023). In the real world, data production is by construction strongly biased: Chemists work with structurally analogous series. In practice, target data is drifting (Tossou et al., 2023), making these models obsolete for real-world production environments. The fundamental factors that influence molecular properties are still unexplored (Deng et al., 2023). With statistical analyses, fixed representations like fingerprints generally match end-to-end deep learning (DL) models in most datasets (Deng et al., 2023; Baptista et al., 2022; Robinson et al., 2020), especially in the presence of activity cliffs (van Tilborg et al., 2022). Much of the progress in molecular property estimation has been driven by the strategic incorporation of inductive biases (Li & Fourches, 2020; Xia et al., 2023a). Techniques such as pretraining (Xia et al., 2023b), the use of equivariant networks, and graph neural networks have proven useful (Satorras et al., 2021; Le et al., 2022). The shift towards leveraging domain-specific knowledge through inductive biases underlines the evolution in our approach to QSAR modeling.

---

[*]Equal contribution    [1]Janssen Research & Development [2]Janssen Pharmaceutica NV. Correspondence to: Jose Arjona-Medina <jarjonam@its.jnj.com>, Ramil Nugmanov <rnugmano@its.jnj.com>.

*Accepted at the 1st Machine Learning for Life and Material Sciences Workshop at ICML 2024.* Copyright 2024 by the author(s).

**Pretraining and Transfer Learning in QSAR Models.**
Pretraining has emerged as a potent solution to overcome data scarcity and enhance model generalizability in various domains, including image and natural language processing. However, without appropriate domain expertise, pretraining can sometimes lead to negative transfer or only marginal performance gains (Rosenstein et al., 2005). In the realm of molecular modeling, pretraining strategies have been successful, particularly when models are pretrained on large-scale datasets (Wang et al., 2019; Chithrananda et al., 2020; Stärk et al., 2023; Xu et al., 2024). For instance, the large-scale pretraining of models has been shown to improve sample efficiency in active learning frameworks used for molecule virtual screening (Cao et al., 2023). Or (Schimunek et al., 2023) which enriches molecule representations with contextual knowledge from reference molecules using a Modern Hopfield Network (Ramsauer et al., 2021). Combining node-level and graph-level pre-training tasks to achieve superior generalization, particularly in out-of-distribution scenarios was presented in (Hu et al., 2020). ChemBERTa utilizes large-scale pretraining datasets to explore the effects of dataset size on downstream task performance (Chithrananda et al., 2020).

**Multitask learning.** Closely related to transfer learning, involves the simultaneous learning of multiple related tasks, enhancing the model's ability to generalize across tasks (Caruana, 1997). This approach has been leveraged effectively in models like MolPMoFiT, which was pretrained using bioactive molecules from ChEMBL for QSPR/QSAR tasks (Li & Fourches, 2020).

**Multimodal and Self-supervised Learning.** Recent advancements have also been made in multimodal learning and self-supervised learning frameworks. The GIMLET model, for example, pretrained on a dataset of molecule tasks with textual instructions, bridging the gap between graph and text data modalities and paving the way for instruction-based pretraining (Zhao et al., 2023). Similarly, the D&D framework utilizes a cross-modal distillation approach, transferring knowledge from 3D to 2D molecular structures, thereby significantly enhancing model performance in downstream tasks (Cho et al., 2023). Or denoising 3D structures as a pre-training objective, setting new benchmarks in the widely used QM9 dataset (Zaidi et al., 2022).

**Contrastive learning** approaches (Le-Khac et al., 2020; Khosla et al., 2020) can overcome data limitations by integrating additional data. For example, CLOOME (Sanchez-Fernandez et al., 2023) embeds bioimages and chemical structures into a unified space, enabling highly accurate retrieval of bioimages based on chemical structures. MolFeS-Cue (Zhang et al., 2024) incorporates a dynamic contrastive loss function tailored for class imbalance. MolCLR (Wang et al., 2022) uses large-scale unlabeled data, graph-based augmentations, and a contrastive learning strategy to improve model generalization. A recent overview of methods for multi-modal contrastive learning is presented in (Seidl, 2024).

Our approach effectively combines pretraining with QM data at node-level, to improve the downstream task of property modeling. We analyze the effect of this pretraining in the distribution of the features of the network, and we observe that atom-level pretrained networks have more Gaussian-like distributions which are more robust to distribution shifts in the input space, establishing a direct connection between pretraining at node-level and supervised learning theory of generalization.

Or contributions can be summarized as follows:

- We empirically show that atom-level pretraining in Graph neural networks with QM properties improves performance over scratch networks and molecular-level pretraining, in the TDC public dataset.

- We empirically show that atom-level pretraining produces a more normal distribution of features, compared to scratch and molecular-level pretrained networks.

- We empirically show that atom-level pretraining produces a more robust molecular representation against distribution shift from train to eval and train to test datasplits, which could explain the performance gain of atom-level pretraining approach.

To the best of our knowledge, this is the first study that aims to understand how atom-level pretraining improves molecular representation.

## 2. Methods

**Graphormer.** In 2021, Microsoft published Graphormer(Ying et al., 2021), a neural network specifically engineered for processing graphs and, more pointedly, molecular structures. This network has exhibited outstanding performance in molecular quantum properties, adapting the architecture of BERT(Devlin et al., 2019) to suit graph data effectively. Graphormer introduces several novel features, the key among them being "centrality encoding". This technique captures the importance of nodes within the graph by integrating graph degree centrality, which is encoded as embedding vectors added into the atom's node type embeddings. Additionally, Graphormer incorporates a "spatial encoding" strategy. This involves representing the shortest path between pairs of nodes, which is then utilized as a learnable bias in the attention matrix. This concept mirrors techniques used in other advanced models like T5(Raffel et al., 2019) and ALIBI(Press et al.,

*Table 1.* Graphormer results

|  | Metric | Direction | scratch | mol-level pretrained HLgap | atom-level pretrained all (4) |
|---|---|---|---|---|---|
| caco2_wang | MAE | ↓ | 0.48 ± 0.06 | 0.53 ± 0.02 | **0.41 ± 0.03** |
| lipophilicity_astrazeneca | MAE | ↓ | 0.58 ± 0.02 | 0.57 ± 0.02 | **0.42 ± 0.01** |
| solubility_aqsoldb | MAE | ↓ | 0.89 ± 0.04 | 0.89 ± 0.02 | **0.75 ± 0.01** |
| ppbr_az | MAE | ↓ | 8.38 ± 0.24 | 8.22 ± 0.23 | **7.79 ± 0.24** |
| ld50_zhu | MAE | ↓ | 0.61 ± 0.02 | 0.60 ± 0.03 | **0.57 ± 0.02** |
| hia_hou | ROC-AUC | ↑ | **0.96 ± 0.03** | **0.96 ± 0.02** | 0.94 ± 0.05 |
| pgp_broccatelli | ROC-AUC | ↑ | 0.87 ± 0.04 | 0.86 ± 0.01 | **0.89 ± 0.02** |
| bioavailability_ma | ROC-AUC | ↑ | 0.52 ± 0.01 | 0.55 ± 0.03 | **0.64 ± 0.05** |
| bbb_martins | ROC-AUC | ↑ | 0.83 ± 0.01 | 0.82 ± 0.03 | **0.88 ± 0.02** |
| cyp3a4_substrate_carbonmangels | ROC-AUC | ↑ | 0.63 ± 0.07 | **0.64 ± 0.03** | **0.64 ± 0.02** |
| ames | ROC-AUC | ↑ | 0.72 ± 0.02 | 0.73 ± 0.01 | **0.80 ± 0.01** |
| dili | ROC-AUC | ↑ | 0.86 ± 0.02 | 0.87 ± 0.01 | **0.88 ± 0.03** |
| herg | ROC-AUC | ↑ | **0.78 ± 0.01** | 0.76 ± 0.04 | 0.77 ± 0.06 |
| vdss_lombardo | Spearman | ↑ | 0.58 ± 0.04 | **0.59 ± 0.04** | **0.59 ± 0.03** |
| half_life_obach | Spearman | ↑ | 0.39 ± 0.07 | 0.34 ± 0.07 | **0.48 ± 0.06** |
| clearance_microsome_az | Spearman | ↑ | 0.49 ± 0.03 | 0.46 ± 0.03 | **0.60 ± 0.01** |
| clearance_hepatocyte_az | Spearman | ↑ | 0.34 ± 0.04 | 0.31 ± 0.02 | **0.46 ± 0.03** |
| cyp2d6_veith | PR-AUC | ↑ | 0.43 ± 0.03 | 0.47 ± 0.02 | **0.61 ± 0.02** |
| cyp3a4_veith | PR-AUC | ↑ | 0.73 ± 0.01 | 0.74 ± 0.03 | **0.80 ± 0.03** |
| cyp2c9_veith | PR-AUC | ↑ | 0.63 ± 0.02 | 0.66 ± 0.03 | **0.69 ± 0.02** |
| cyp2d6_substrate_carbonmangels | PR-AUC | ↑ | 0.52 ± 0.01 | 0.54 ± 0.04 | **0.58 ± 0.03** |
| cyp2c9_substrate_carbonmangels | PR-AUC | ↑ | 0.35 ± 0.02 | 0.33 ± 0.03 | **0.37 ± 0.04** |

*Figure 1.* Different aromatic ring representations folded into a single form

2021), highlighting its relevance and utility in enhancing the model's attention mechanism within graph networks.

**Custom implementation.** Following (Nugmanov et al., 2022), we have refined the "centrality encoder" of the original Graphormer to encompass not just explicit neighbors but total neighbors, integrating both explicit atom neighbors and implicit hydrogens. This approach eliminates the need for an "edge encoder", leveraging a combination of the centrality encoder and an atom type encoder to implicitly capture atom hybridization. Moreover, we have streamlined the model by omitting the encoding of atoms' formal charges and radical states, allowing for the representation of resonance structures in a unified form and dispensing with the traditional concept of "aromatic" bonds typically applied to arenes. Additionally, our model introduces a "spatial encoder" constrained by a configurable maximum distance threshold to enhance computational efficiency and model accuracy. By treating distances beyond this threshold uniformly, the model focuses on more relevant short-range

interactions understanding molecular structures.

**Enabling multitask learning by task-specific virtual nodes.** Similar to the original BERT model, Graphormer utilizes a [CLS] token as a global readout for estimating properties of graphs and molecules. This methodology aligns with techniques used in graph convolutional networks (GCNs), where virtual nodes connected to every node of the graph can serve a comparable purpose (Cai et al., 2023). Building on this concept, we propose a novel extension tailored for multitask learning applications. In our approach, we employ the same encoder and output head to estimate multiple properties, but differentiate the tasks by using distinct virtual nodes for each. This allows for task-specific processing while maintaining a unified architecture, enhancing the model's efficiency, adaptability, and scalability in handling diverse learning tasks.

**Quantitative Assessment of learned features.** To analyze the impact of pretraining methods on the molecular

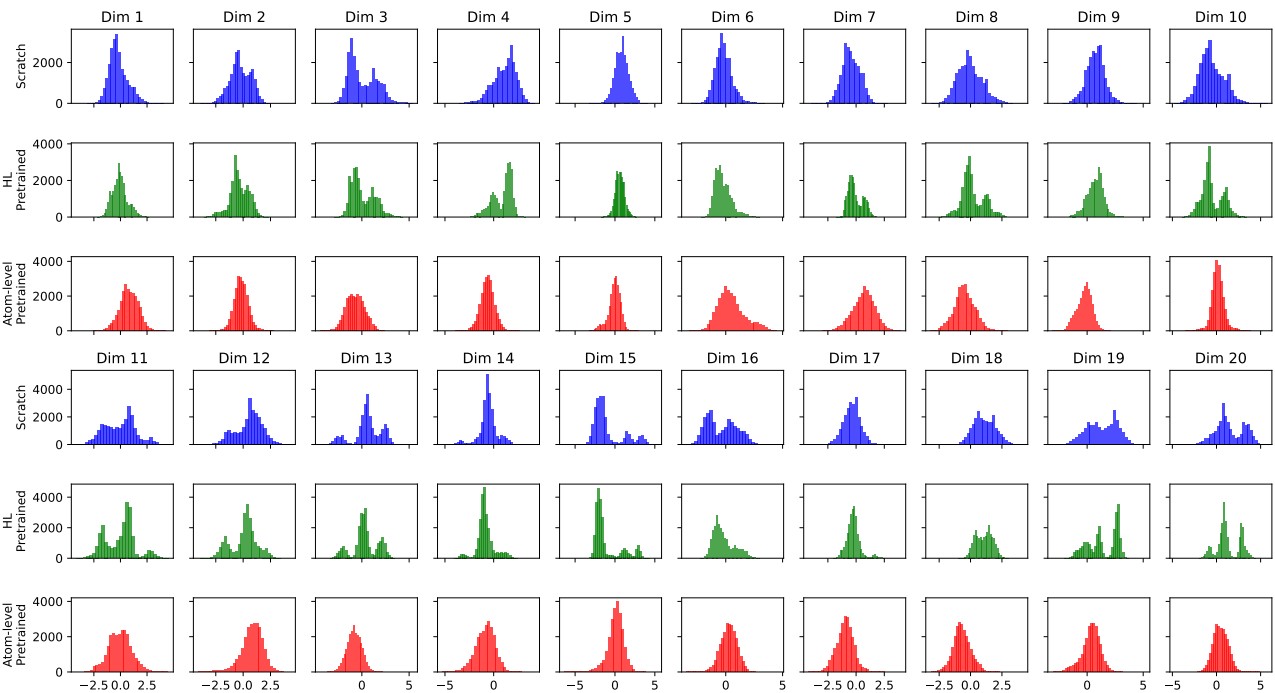

Distribution of Activations for 20 Features

*Figure 2.* Distribution of first 20 features from the first layer of the Graphormer network for three different training approaches —scratch, HOMO-LUMO pretrained and atom-level pretrained— across test split of lipophilicity dataset .

representation captured by the model, we conducted a comprehensive analysis of learned feature distributions. We run the models on the three different data splits (training, validation, and test), and extract the distribution of features after the first layer of the Graphormer architecture.

For each compound in the dataset, we aggregated the node representations after the first Graphormer layer, resulting in a cumulative dataset-wide node representation. This aggregation facilitated the derivation of an empirical distribution of activations per model and data split. Using this empirical distribution, we applied various statistical techniques to quantify the differences in feature distributions across the different data splits for each of the models. Furthermore, we undertook an evaluation of the normality of these distributions using Shapiro-Wilk and Kolmogorov-Smirnov tests. This assessment was used to determine whether significant deviations exist in the distribution of learned features across different pretraining methods.

## 3. Experiments

**Datasets** For pretraining, we used a publicly available dataset (Guan et al., 2021) consisting of 136k organic

molecules. Each molecule is represented by a single conformer generated using the Merck Molecular Force Field (MMFF94s) in the RDKit library. The initial geometry for the lowest-lying conformer was then optimized at the GFN2-xtb level of theory followed by refinement of the electronic structure with DFT (B3LYP/def2svp). The advantage of the described dataset is several reported atomic properties: charge, electrophilicity, nucleophilicity Fukui indexes, and an NMR shielding constant. Some additional curation was carried out as described in the addendum. Another pretraining dataset, PCQM4Mv2, consists of a single molecular property per molecule, a HOMO-LUMO gap https://ogb.stanford.edu/docs/lsc/pcqm4mv2/. It was curated under the PubChemQC project (Nakata & Shimazaki, 2017). For the benchmarking of the obtained pretrained models, we used the absorption, distribution, metabolism, excretion, and toxicity (ADMET) group of the Therapeutics Data Commons (TDC) dataset (Huang et al., 2022), consisting of 9 regression and 13 binary classification tasks for modeling biochemical molecular properties.

**Experiments setup** We train our Graphormer network from scratch on the set of tasks provided by the TDC dataset. We also pretrained the same network using as main task

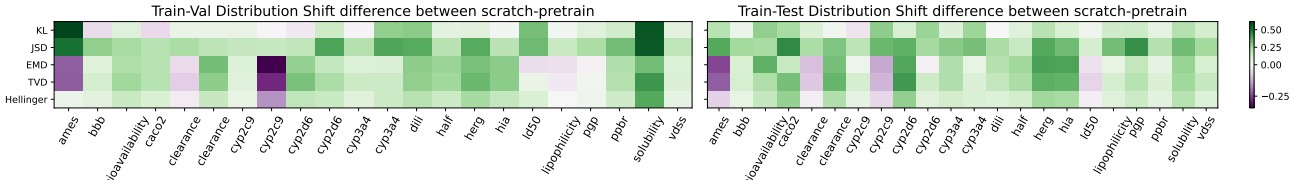

*Figure 3.* This heatmap illustrates the differences in distribution shifts between scratch and pretrained networks, calculated across various feature dimensions using metrics such as Kullback-Leibler Divergence, Jensen-Shannon Divergence, Earth Mover's Distance, Total Variation Distance, and Hellinger Distance. Green hues indicate instances where the pretrained network exhibits smaller distribution shifts compared to the scratch network, while purple hues denote the opposite. Notably, both heatmaps comparison frequently show reduced distribution shifts in pretrained networks, specially for Train-Test comparison, which likely helps to explain pretrained network's superior performance on the test data split.

HOMO-LUMO gap (molecule-level) and the atomic properties charge, electrophilicity, and nucleophilicity Fukui indexes and an NMR shielding constant (atom-level). For atom-level pretrain, we pretrained 4 models, one per every single property. We also pretrained a fifth model which is pretrained on 4 properties via multi-task by task-specific virtual nodes. Main results reported in this paper for atom-level uses this last model. Results for specific atomic property pretrained are available in the Supplementary Material.

## 4. Results

This section provides a concise summary of the key findings of this study. For an exhaustive elaboration of the results, please refer to the supplementary materials.

**Pretraining on atom-level node with QM significantly improves performance in downstream tasks**   In Table 1, we present the outcomes from benchmarking three distinct training approaches: scratch, molecule-level QM pretrained, and atom-level QM pretrained with all properties for 5 different seeds, as described in the guidelines provided by the TDC dataset https://tdcommons.ai/benchmark/overview/. We have excluded the results for atom-level pretraining on individual QM properties from this table; these can be found in the Supplementary Materials. These results show that atom-level pretraining notably enhances model performance compared to training from scratch for 21 of the 22 datasets.

**Pretraining on atom-level node with QM results on smoother feature distribution**   We further examined the distribution of the features from the network's first layer for atom-level pretrained, molecular-level (HOMO-LUMO gap) and scratch models across different data splits and datasets. Figure 2 illustrates the feature distribution for the test split of one of the datasets (lipophilicity), highlighting how scratch and molecular-level pretrain compare to the atom-level pre-

trained network (for other datasets and splits, please refer to the Supplementary Materials). Visually, we can observe that the atom-level pretrained method results in a more Gaussian-like distribution of the features. We conducted a Shapiro-Wilk test for each dimension in both scratch and pretrained networks to assess the normality of the distributions. The average p-value for the scratch network was 3.2E-06 with a standard deviation of 5E-05, suggesting a significant deviation from normality. In contrast, the pretrained network had an average p-value of 4E-04 with a standard deviation of 0.003, indicating a closer approximation to a Gaussian distribution.

**Pretraining on atom-level node with QM enhances robustness to input distribution shifts**   Pretrained networks show in general less distribution shift compared to scratch network, as we can see in Figure 3. In this figure, we show the distribution shifts differences between scratch and pretrained networks. As we can appreciate, in most of the cases, specially in the Train-Test comparison, atom-level pretrained networks have less distribution shift than scratch networks.

## 5. Conclusions

In this study, we have demonstrated that pretraining of graph-based neural networks with atom-level quantum mechanics (QM) data significantly enhances performance on downstream tasks related to ADMET properties within the TDC dataset, as illustrated in Table 1. Furthermore, we showed the change in the distributions of activations of the internal model's features due to specific pretraining. After atom-level pretraining with QM data, these distributions become more Gaussian-like, which is known to be conducive to better learning dynamics and thus improved performance (see Figure 2). Moreover, our findings indicate that pretrained models exhibit smaller distribution shifts from training to testing datasets, further supporting the efficacy of QM data

pretraining in enhancing model robustness (see Figure 2).

To our knowledge, this is the first study that elucidates how atom-level pretraining can optimize molecular representations by analyzing the model's internal representation and robustness to distribution shifts.

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
