# Analysis of Atom-level pretraining with QM data for Graph Neural Networks Molecular property models

## Abstract

Despite the rapid and significant advancements in deep learning for Quantitative Structure-Activity Relationship (QSAR) models, the challenge of learning robust molecular representations that effectively generalize in real-world scenarios to novel compounds remains an elusive and unresolved task. This study examines how atom-level pretraining with quantum mechanics (QM) data can mitigate violations of assumptions regarding the distributional similarity between training and test data and therefore improve performance and generalization in downstream tasks. In the public dataset Therapeutics Data Commons (TDC), we show how pretraining on atom-level QM improves performance overall and makes the activation of the features distributes more Gaussian-like which results in a representation that is more robust to distribution shifts. To the best of our knowledge, this is the first time that hidden state molecular representations are analyzed to compare the effects of molecule-level and atom-level pretraining on QM data.

# A   Supplementary Materials

## A.1   Distribution of activations of features for scratch, HOMO-LUMO pretrained and Atom-level pretrained networks in TDC Lipophilicity Dataset

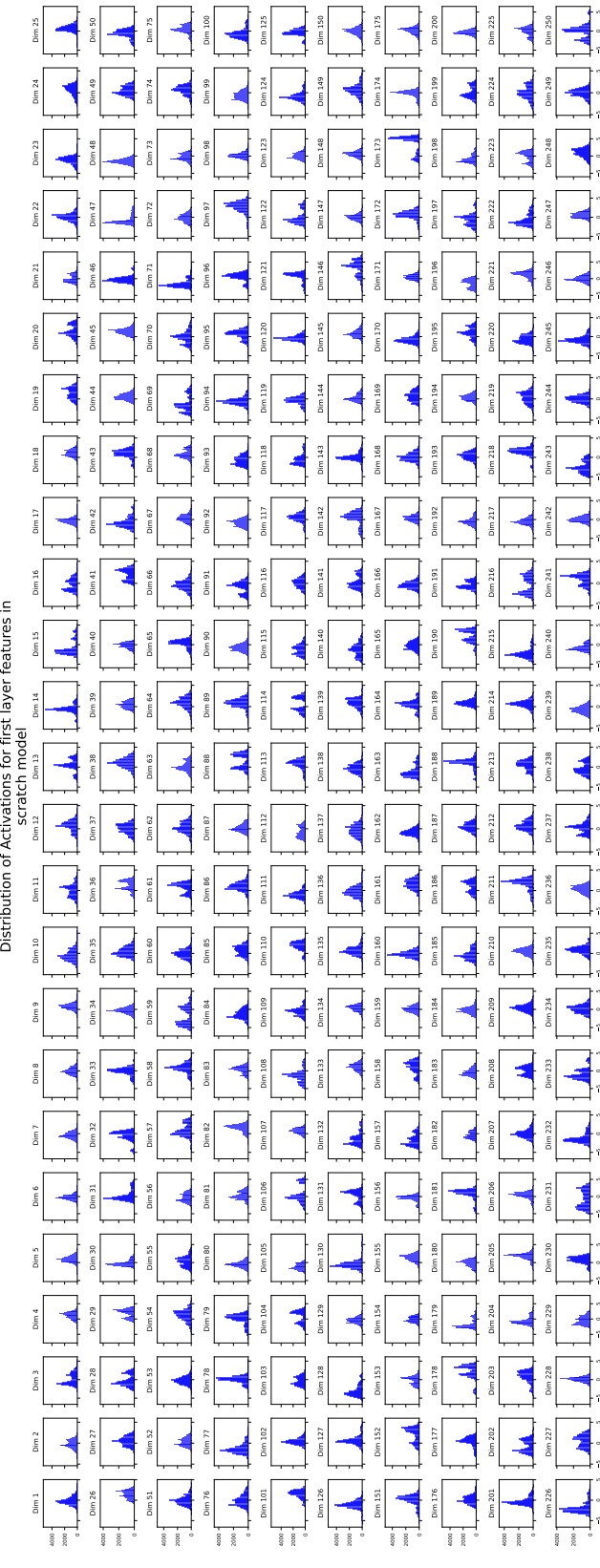

Figure 1: Distribution plots Scratch for Lipophilicity test set

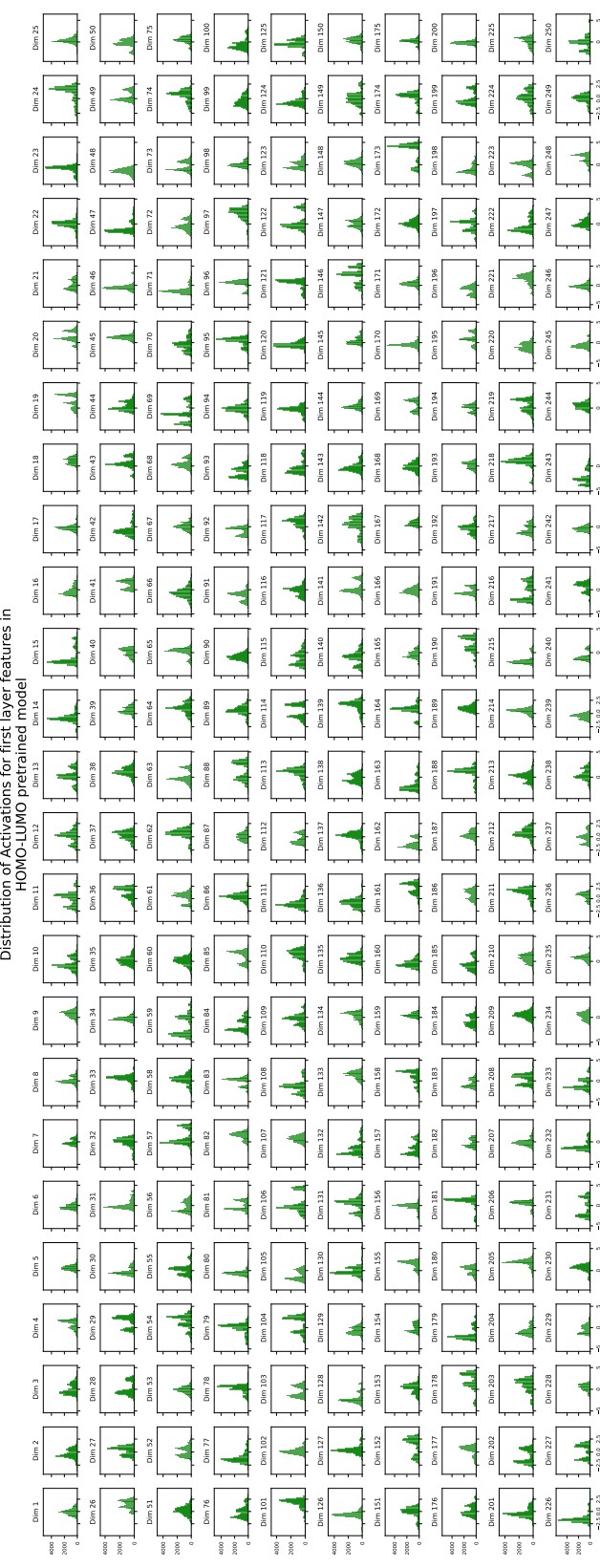

Figure 2: Distribution plots HOMO-LUMO pretrained for Lipophilicity test set

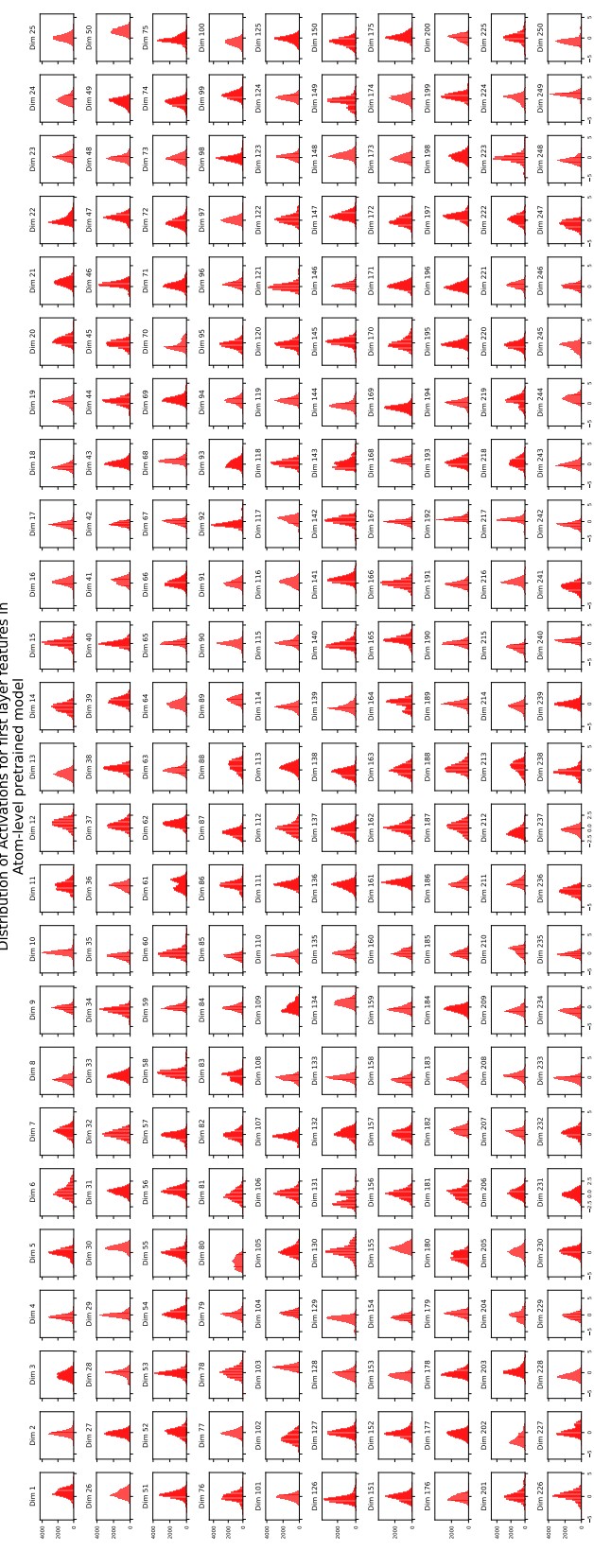

Figure 3: Distribution plots Atom-level pretrained for Lipophilicity test set

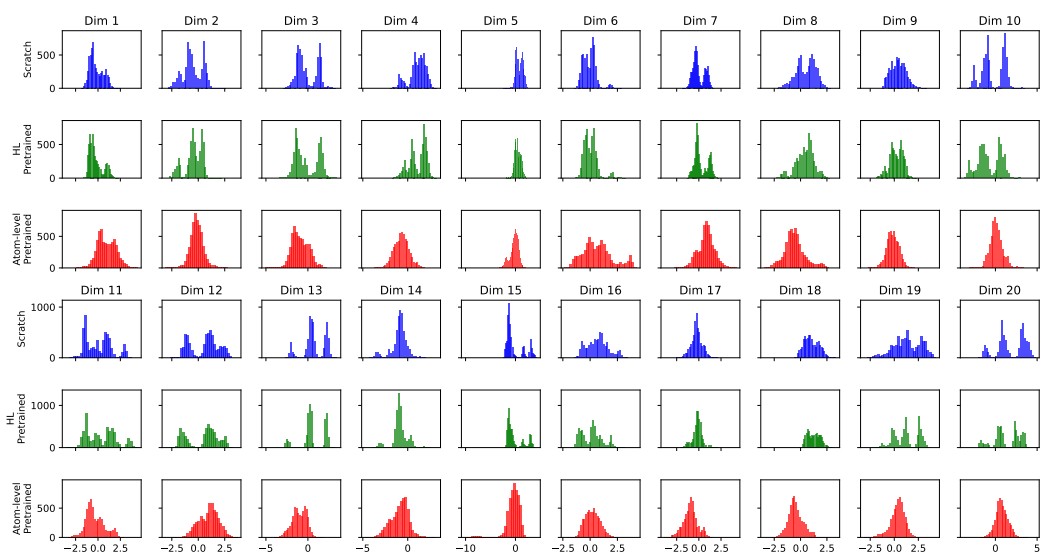