# OpenReview forum: "Analysis of Atom-level pretraining with QM data for Graph Neural Networks Molecular property models"
_ICML.cc/2024/Workshop/ML4LMS — ML4LMS Poster_

### Official Review · Reviewer_HZ59 · 2024-06-03
**The manuscript is well-written and clear, but presents results of marginal significance. Overall, it is above acceptance threshold and could be improved on a few points.**

**Rating:** 6
**Confidence:** 3

**Review:**

The manuscript analyses how pretraining quantitative structure-activity relation (QSAR) models with quantum mechanical (QM) atomic information improves performance and generalization on downstream tasks. Furthermore, it empirically shows how pre-training makes the feature distributions more Gaussian in the first layer of a graph transformer (Graphormer) model.

The manuscript is well-written and clear, and cleanly states the problem it is trying to analyse and the benchmarks it wants to improve upon. The idea of pre-training on atomic QM data to improve generalizability on other prediction tasks is, as noted by the authors, not original, but the manuscript contains an additional analysis of the model weights which provides additional insight into the mechanism behind the improvements in model performance offered by pre-training.

The manuscript’s significance is, in my opinion, marginal, as it does not present a novel methodology nor it showcases dramatic improvements on a wide array of benchmarks.

Overall, the manuscript satisfies the criteria for acceptance in my view. I would suggest the authors to improve on the following points to further increase the clarity and impact of their present work:

1) In Lines 118-120, left column, the authors state that "Gaussian-like distributions [...] are more robust to distribution shifts in the input space". It is not clear to me whether this is an empirical observation the authors make after analysing their results, or if it is a well-known result. In the former case, the authors should make it more explicit that this is an empirical claim of their manuscript, in the latter case the authors should provide one or more citations to support their statement. Moreover, if the statement is indeed a claim made by the authors in view of the results they present, a more in-depth analysis is required to understand whether there is a significant correlation between weight distributions being Gaussian and train-test distribution shifts being smaller.

2) Figure 1 is never referenced in the main text, this should be fixed

3) Labels in the x-axis of Figure 3 are often repeated. The authors should fix this and map the labels 1-to-1 with the entries in Table 1. I would also suggest to maintain the same order of labels to improve legibility.

4) In Figure 3,  given how the authors are plotting a quantity that, in principle, should be symmetric around 0, they could consider shifting the colourmap to be comprised between -0.6 and +0.6. This would have the added benefit of better highlighting the reduced distribution shifts in pretrained networks.

5) Line 94, left column, starts a sentence with “Or…”, the syntax could be improved.

6) The authors should report the p-value and standard deviations for the Shapiro-Wilk test also for the case of HL-pretrained networks, for completeness.

7) The authors state, in line 250, left column “For an exhaustive elaboration of the results, please refer to the supplementary materials”. This sentence could deceive the reader, since the Supplementary Material currently contains only 25 Figures with no other text and/or analysis of the results. Please either amend the sentence or, preferably, include further analysis of the results in the Supplementary Material.

---

### Official Review · Reviewer_dXDJ · 2024-06-11
**Analysis of the impact of molecule-level and atom-level pretraining on hidden-state molecular representations**

**Rating:** 8
**Confidence:** 3

**Review:**

The Authors pretrain their custom implementation of Graphformer
on both atom-level and molecule-level properties
and benchmark the models on the ADMET group of TDC.
They show that the atom-level pretraining improves the performance
and makes the features’ distribution more Gaussian-like.

Questions and comments:

1) Stating early in the introduction and in the methods that this work focuses on 2D molecular graphs
(probably from SMILES?) might improve clarity of the paper.

2) Do I understand correctly that the dataset of [Guan et al., 2021] was used for pretraining
on atom-level properties and the PCQM4Mv2 was used to pretrain on HOMO-LUMO gap?
Can the differences in pretrained models’ performance arise from this?

3) On Figure 2, the HOMO-LUMO pretrained model features look “more” bimodal than the scratch
model. Is it possible that the target distribution of the pretraining set is bimodal? It would
be interesting to see the pretraining and downstream target properties distributions along with
the feature distributions. Is Gaussian distribution of features always good (especially in case of
activity cliffs)?

4) Figure 3 shows the distribution shifts differences between scratch and pretrained networks.
Which pretrained network was used, the atomic properties one? Is small distribution shift always good
considering data drift?

5) In Table 1, the upstream task results are reported and take the larger part of the table, but are not discussed. Also, for readers unfamiliar with the datasets used, it would be useful to extend the table caption.

6) What is Figure 1 for?

7) Minor: there are quite a few typos (e.g. “makes the activation of the features distributes more Gaussian-like” in the abstract, no full stop in the end of the second paragraph of the introduction, strange punctuation in the paragraph titles, spacing before citations, quotation marks typesetting, “Experiments setup here a change”, “3.2E-06”, “shift than scratch networks. T”).